behaviour

bats, biologging, habituation, sensor, telemetry, tracking

**Author for correspondence:**
Gerald G. Carter
e-mail: carter.1640@osu.edu

# Habituation of common vampire bats to biologgers

Emma Kline[1], Simon P. Ripperger[1,2,3] and Gerald G. Carter[1,3]

[1]Department of Evolution, Ecology and Organismal Biology, The Ohio State University, Aronoff Laboratory, 318 W 12th Avenue, Columbus, OH 43210, USA
[2]Museum für Naturkunde–Leibniz Institute for Evolution and Biodiversity Science, Berlin, Germany
[3]Smithsonian Tropical Research Institute, Ancón, Republic of Panama

 SPR, 0000-0003-1527-8657; GGC, 0000-0001-6933-5501

Rapid advancements in biologging technology have led to unprecedented insights into animal behaviour, but testing the effects of biologgers on tagged animals is necessary for both scientific and ethical reasons. Here, we measured how quickly 13 wild-caught and captively isolated common vampire bats (*Desmodus rotundus*) habituated to mock proximity sensors glued to their dorsal fur. To assess habituation, we scored video-recorded behaviours every minute from 18.00 to 06.00 for 3 days, then compared the rates of grooming directed to the sensor tag versus to their own body. During the first hour, the mean tag-grooming rate declined dramatically from 53% of sampled time (95% CI = 36–65%, $n = 6$) to 16% (8–24%, $n = 9$), and down to 4% by hour 5 (1–6%, $n = 13$), while grooming of the bat's own body did not decline. When tags are firmly attached, isolated individual vampire bats mostly habituate within an hour of tag attachment. In two cases, however, tags became loose before falling off causing the bats to dishabituate. For tags glued to fur, behavioural data are likely to be impacted immediately after the tag is attached and when it is loose before it falls off.

## 1. Introduction

Recent and rapid advancements in animal-borne telemetry devices (biologgers) have allowed for unprecedented insights into animal behaviour in the wild, especially for animals that are difficult to observe like bats [1–5]. Despite the success and expansion of biologging, guidelines for their proper and ethical use might not generalize to different devices or species, posing potential problems for data or animal welfare [6–8]. Several studies that involve animal tagging explicitly aim to quantify the effects of the deployed animal-borne tags on the subjects' behaviour [9–11]. A meta-analysis of 37 bird tagging studies revealed a small but significant increase in foraging trip

duration [10], and there is increasing evidence for species- and behaviour-specific tagging effects in mammals. For example, tagged bottlenose dolphins (*Tursiops truncates*) swam 11% slower than non-tagged control animals [12], Eurasian beavers (*Castor fiber*) responded with small but clear changes in activity levels when tagged [13], and grazing in plains zebra (*Equus burchelli antiquorum*) appeared to be influenced by GPS collar size [14].

A central goal of most biologging studies is to collect accurate, unbiased data that represent 'typical' behaviour. Researchers typically assume that animals will either not respond or quickly habituate to biologgers. However, biologgers that are attached to an animal's body can cause irritation and changes in activity when it expends time and energy trying to remove the tag until it habituates to the logger's presence. Social animals that are tagged may also reduce their social interactions, and consequently, tagging may lead to underestimation of social interaction rates until habituation occurs. The same is true for visual observations of animals not habituated to the presence of a human observer, but this problem is obvious to the researcher.

Here, we measured how quickly wild-caught, isolated, captive common vampire bats (*Desmodus rotundus*) habituated to proximity sensors that are glued to their dorsal fur [5,15]. These lightweight sensors allow remote tracking of pairwise social encounters. They are glued to the fur temporarily and typically fall off on their own after 1–2 weeks. To measure habituation, we estimated the amount of time bats spent scratching at the tag (tag-grooming rate) versus grooming other parts of their body (self-grooming rate).

## 2. Material and methods

To test if and how bats habituate to attached proximity tags, we first constructed mock proximity tags that were identical in mass (1.5 to 1.8 g) and shape to real proximity loggers [5,15] using the same plastic housing and antenna wire. We used 'Osto-Bond skin bonding latex adhesive' (Montreal Ostomy Products, Vaudreuil, Quebec, Canada) to attach the devices to dorsal fur in the typical way, i.e. applying glue to the tag and to the fur on the bat's back, slightly lower than their shoulder blades, letting the glue dry for 1–2 min and then placing the tag on this spot once the glue became tacky [15]. To observe individual behaviour, we placed individual bats in clear plastic cages ($28 \times 28 \times 40$ cm) with plastic mesh along the top and one side for roosting, and we used infrared home-surveillance cameras to record the top half of the cage from 18.00 to 06.00 for 3 days. Although the bats could not fly, our goal here was to measure how the tags would affect the behaviour of roosting bats. During daytime, we covered the cages with dark fabric. Every evening between 18.00 and 23.00, we fed bats with cow blood defibrinated with sodium citrate (11 g of sodium citrate and 4 g of citric acid per 4 l of blood collected from a slaughterhouse).

Subjects were 13 adult common vampire bats captured from three distant sites in Panama and housed at the Smithsonian Tropical Research Institute (STRI) in Gamboa, Panama for the duration of the experiment. We tested the bats from each site in three blocks of time. For block 1 (6 June 2019), we tested five males that were captured from a cave at Lake Bayano, then individually banded with a 3–4 mm forearm band on 5 May. For block 2 (16 June), we tested five females that were captured from a tree roost in Tolé and banded on 8 June. For block 3 (23 June), we tested three females from a tree roost in La Chorrera that were captured on 13 June and banded on 19 June. Before the bats were tagged and housed in clear plastic cages, they were housed communally in a flight cage (either 2.1 × 1.7 × 2.3 m or approx. 1.7 × 1 × 2.3 m).

For every minute from 18.00 to 06.00, we scored the presence or absence of self-grooming (scratching or licking the body) and tag-grooming (scratching, pulling, or attempting to bite the proximity logger case or wire). We assessed habituation as a decrease in tag-grooming rates, compared to self-grooming rates, over time. We rounded the time of observations to the nearest hour (e.g. 0 h is time 0 to 00.29.59, hour 1 is 00.30.00 to 1.29.59), then used non-parametric bootstrapping to generate a 95% confidence interval around the mean rate of self-grooming and tag-grooming for every hour after attachment. For bootstrapping, we used the percentile method in the boot R package [16–18].

When estimating grooming rates, we excluded minutes sampled when the bat was out of the camera's view (17%) or responding to a human in the room (1.8%). Unfortunately, video data for the first 2 h after attachment were lost for four of 13 bats total (four of five bats in time block 2). Samples per bat ranged from 1689 to 2346 except for one bat that removed the tag early (sampled only 626 times).

This work was approved by the Smithsonian Tropical Research Institute Animal Care and Use Committee (no. 2015-0501-2022) and the Panamanian Ministry of the Environment (no. SE/A-67-2019).

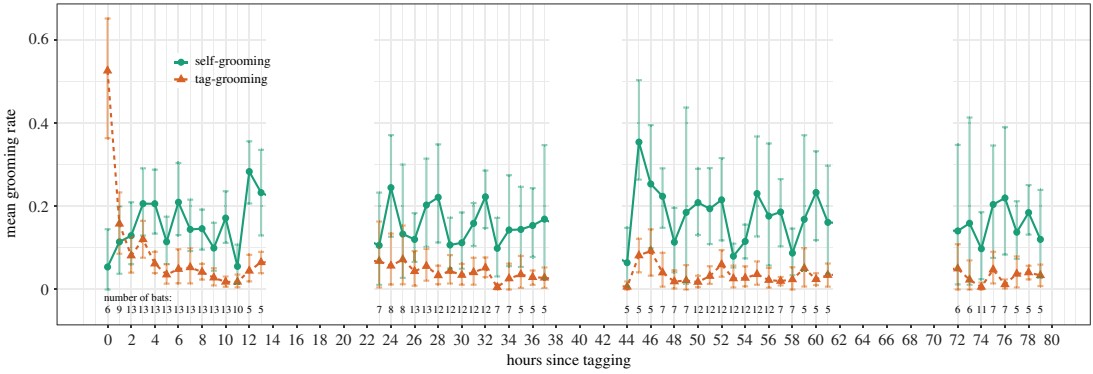

**Figure 1.** Mean grooming rates over time. Mean rates of grooming directed at the proximity sensor tag (orange triangles and dashed line) versus rest of the body (green circles and solid line). To create time bins, observations were rounded to the nearest hour. Error bars show bootstrapped 95% confidence intervals. To exclude imprecise grooming rates, we only show mean rates for hours with data from more than three bats.

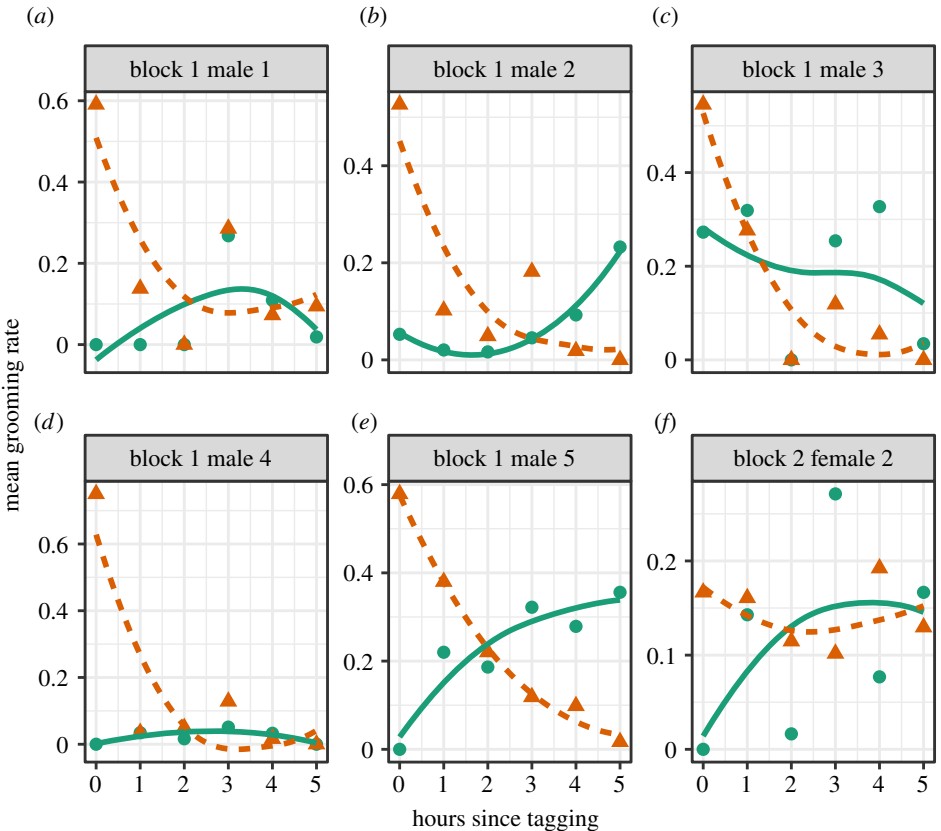

**Figure 2.** Habituation during first 5 h for the six bats ((*a*) block 1 male 1, (*b*) block 1 male 2, (*c*) block 1 male 3, (*d*) block 1 male 4, (*e*) block 1 male 1 and (*f*) block 2 female 2) that were observable on camera within 30 min of tagging. Curves are fitted rates of grooming directed at the proximity sensor tag (orange triangles and dashed line) versus rest of the body (green circles and solid line) based on local polynomial regression fitting (span = 2) in the ggplot2 R package [19,20]. To create time bins, observations were rounded to the nearest hour.

## 3. Results

During the first hour, the mean tag-grooming rate declined dramatically from 53% (95% CI = 36–65%, $n = 6$ bats) to 16% (8–24%, $n = 9$ bats; figure 1). From hours 4 to 79, the mean tag-grooming rate ranged from 0 to 9.3% (mean = 3.6%, 95% CI = 3.0–4.2%, $n = 51$ h; figure 1). During this same period of time, mean grooming rates directed to other places on the bat's body did not decline (figure 1). The habituation was evident in tag-grooming rates in comparison to self-grooming rates within the first hour across all six bats that were observable during that time (figure 2). Seven other bats were not

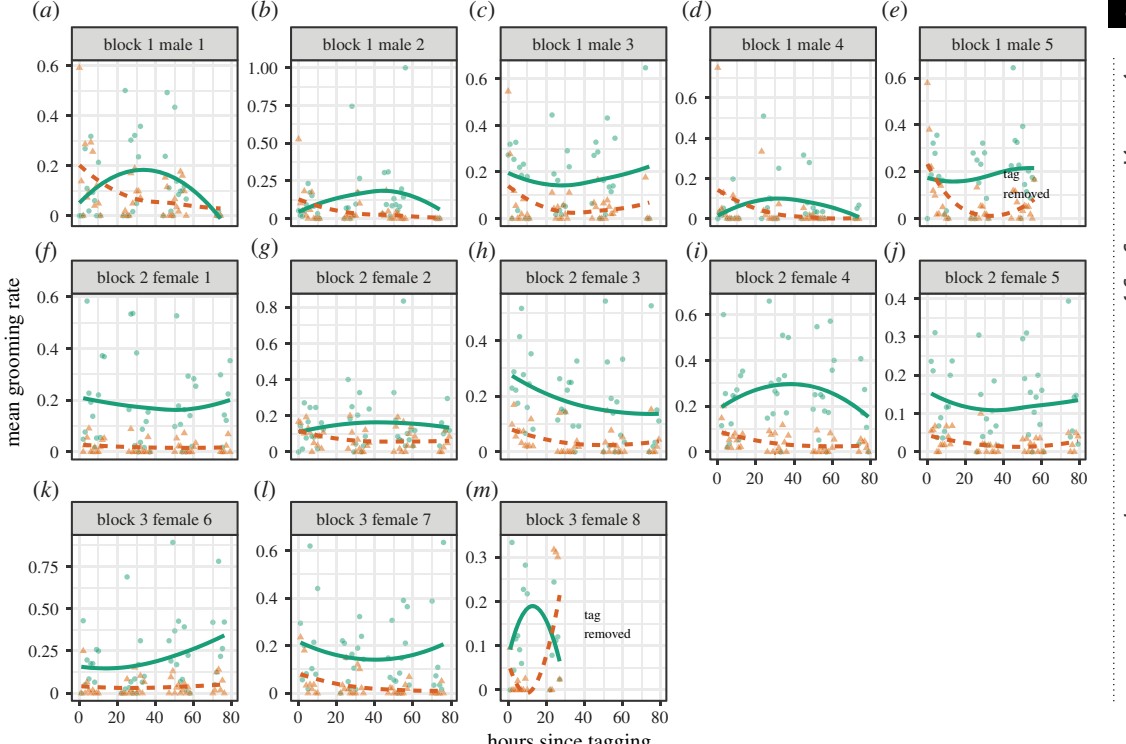

**Figure 3.** Dishabituation in bats that removed loose tag. Lines show fitted curves of grooming directed at the proximity sensor tag (orange dashed line) versus rest of the body (green solid line) based on local polynomial regression fitting (span = 2) in the ggplot2 R package [19,20]. To create time bins, observations were rounded to the nearest hour. Two bats (male 5 and female 8) removed their tags (where the observations end); (a) block 1 male 1, (b) block 1 male 2, (c) block 1 male 3, (d) block 1 male 4, (e) block 1 male 5, (f) block 2 female 1, (g) block 2 female 2, (h) block 2 female 3, (i) block 2 female 4, (j) block 2 female 5, (k) block 3 female 6, (l) block 3 female 7 and (m) block 3 female 8.

filmed (e.g. off-camera and possibly exploring the floor of the cage) during the first hour. In all bats, however, the tag-grooming rates either decreased or remained low after hour 4 (figure 1), with the exception of two bats that dishabituated after they began to remove their tag (figure 3).

## 4. Discussion

Solitary tagged vampire bats habituated to glued-on tracking devices within a few hours as long as the device remained securely attached. The bats spent more than half their time manipulating the tag during the first half-hour, but by the fifth hour, tag-grooming rates dropped to an average of 4% and were lower than self-grooming rates (directed to the bat's body). Self-grooming rates remained stable over time (figure 1; mean = 16%; 95% CI = 14 to 18%, range = 0 to 53%; $n = 58$ h) and were comparable with estimates from other captive studies (12%, [21]) and field observations (24%, [22]).

It is important to note the limitations of this study. We measured a single behaviour in a few isolated individuals. Although we compare tag-grooming with self-grooming as a control measure, we do not have a control period prior to the application of the tag. We also cannot know how these tags affect behaviours such as flight performance or social behaviour. One should therefore be cautious overgeneralizing from these findings. However, some observations are noteworthy. First, two bats were able to loosen the attachment of the tag which caused them to dishabituate and increase tag-grooming rates. The behaviour of tagged animals might therefore be impacted both immediately after tag attachment and immediately before tags fall off. Second, these tags fell off much faster than in a previous study where all 50 proximity sensors that were attached to free-ranging vampire bats stayed attached for 9 days or longer [15]. In this earlier study, we used a different surgical glue (the rubber-based Perma-Type Surgical Cement, Perma-Type Company, Inc., Plainville, Connecticut) which is almost twice as strong as the latex-based surgical glue used here, Osto-Bond [23]. When using glue

for tag attachment, we therefore recommend species-specific tests with different glues [23] and reporting what glue is used.

An important next step is to evaluate to what extent animal-borne tags alter social behaviours. Social effects are likely to vary greatly by species and behaviour; captive African elephants (*Loxodonta africana*) showed no changes in social behaviour when wearing tracking collars [24], but free-ranging greater spear-nosed bats (*Phyllostomus hastatus*) showed increased aggression towards individuals with glowing tags compared to untagged control bats, possibly due to neophobia or light aversion [25]. If proximity sensors alter association rates among tagged pairs for a period after attachment, then this should be taken into account when constructing social networks.

There is increasing evidence across animal taxa that tagging can have considerable effects on survival, reproduction, parental care or behaviour [10]. In insectivorous bats that forage in flight, there is experimental evidence that tagged individuals experience a decline in manoeuvrability [26], but no effects on reproduction or body mass from radio-tagging were detected in a long-term field study [27]. Changes in body mass before and after tagging are common ways of evaluating possible tagging effects, but sample sizes are often small and hence incapable of detecting subtle effects [28,29]. Also, a sampling bias occurs if not all tagged animals are recaptured for measurement, because the animals that are most impacted might not be measured. For these reasons, captive measurements of tagging effects provide an important method for estimating behavioural responses and are useful for biologging studies.

## 5. Conclusion

In solitary vampire bats, habituation to proximity sensors occurs mostly within the first 1–3 h of tag attachment—but only if the tag is securely attached. Attachment methods should therefore be carefully considered and tested. Our findings highlight the need for preliminary testing of biologgers and other tags with captive animals whenever feasible.

Ethics. This work was approved by the Smithsonian Tropical Research Institute Animal Care and Use Committee (no. 2015-0501-2022) and the Panamanian Ministry of the Environment (no. SE/A-67-2019).

Data accessibility. Data and R code are provided as electronic supplementary material [30].

Authors' contributions. E.K. scored the videos to collect data and participated in data analysis, study design and writing; S.P.R. helped with data collection and writing; G.G.C. participated in data analysis, study design and writing. All authors gave final approval for publication and agree to be held accountable for the work performed therein.

Competing interests. We declare we have no competing interests.

Funding. E.K. was supported by the Second Year Transformational Experience Program at The Ohio State University. Work by G.G.C. is supported by the National Science Foundation (Integrative Organismal Systems) under grant no. 2015928.

Acknowledgements. We thank Rachel Page, Gregg Cohen, Sebastian Stockmaier, Imran Razik and Bridget Brown for help with animal care, permits and logistics. Two anonymous reviewers provided valuable feedback on the manuscript.

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
