## [Peer Review File · Royal Society Open Science]

Review History

RSOS-211249.R0 (Original submission)

Review form: Reviewer 1

Is the manuscript scientifically sound in its present form?

No

Are the interpretations and conclusions justified by the results?

Yes

Is the language acceptable?

Yes

Do you have any ethical concerns with this paper?

No

Have you any concerns about statistical analyses in this paper?

No

Recommendation?

Reject

Comments to the Author(s)

Comments on the manuscript entitled: Habituation of common vampire bats to biologgers

In the well-written manuscript, the authors present a test of habituation to tagging in common vampire bats. They assessed the tagging effect by video recording grooming behaviour shortly after the tag deployment. Overall, I like the idea of bridging the short- (rarely studied) and long-term (well studied) effects of biologging. Their results provide a valuable first insight into the process of habituation and might be helpful to researchers planning to start deploying tags to bats. I provide a couple of comments/suggestions that might improve the current version of the manuscript.

Main comments:

- 1) Authors used one type of tag on one species with very specific behaviour using rather low number of wild-caught individuals kept in cages. All of these aspects might have effects on the recorded behaviour and further generalization of the results in any research. However, any of these crucial aspects limiting generalization of manuscript results is neither discussed in the discussion nor mentioned anywhere in the manuscript. I would suggest to properly acknowledge these circumstances at least in the Discussion section.
- 2) Please specify conditions (L103-110) under which you kept bats after the bats were captured and before the bats were tested (approx. 1 month). Did you keep them in the small plastic boxes?
- 3) You state (L96) the bats were caged in small (28×28×40 cm) plastic boxes. I assume that bats could not fly in such small boxes and their behaviour was thus limited. This is another factor that could affect the tagging effects as bats could not behave naturally but is not discussed in the manuscript.

####

Minor comments:

- L29: Could you specify (daytime/night) period of the day when individual behaviour was recorded? Was the behaviour recorded continuously for 12 hours, or over the 12-hour period in total?
- L33: Please state sample sizes for these results if different from number of individuals used in the study (please see comment to L123-124 below).
- L38: What does "solitary" means in this sentence? Is it related to species name, caging of individuals, something else?
- L41: I would suggest to replace "remove" with "fall off".
- L70-71: I would say that tags that are attached to animal's body with a harness can cause irritation too.
- L79: You stated that 13 individuals were tested multiple times (L25, L103, Figure 3).
- L122: What is a "flight cage"? This was not described yet.
- L123-124: Therefore, short-term habituation could be assessed in nine individuals (assuming you had 13 individuals, not 14). This should be clearly stated throughout the manuscript.
- L132: Please provide sample size for each result you provide - this will (i) allow easy interpretation of your results and (ii) allow incorporation of your results into future meta-analysis summarizing tagging effects across species.
- L137: Do you really present relative measures of grooming? It seems to me you present only raw measures of tag- and self-grooming times.
- L204: Did you mean "tracking" instead of "tacking"?
- Figure 1: Please remove lines connecting measures within the 12-hour periods as no values have been measured during these periods.
- L218-219: The effect of glue types was not tested in this study.
- L230-240: I would suggest to remove the part of the text where authors elaborate on the anecdotal observations that are out of scope of their study.

Review form: Reviewer 2

Is the manuscript scientifically sound in its present form?

No

Are the interpretations and conclusions justified by the results?

No

Is the language acceptable?

Yes

Do you have any ethical concerns with this paper?

No

Have you any concerns about statistical analyses in this paper?

Yes

Recommendation?

Major revision is needed (please make suggestions in comments)

Comments to the Author(s)

The manuscript is an interesting contribution to the literature regarding the effects of animal-tagging. Some recommendations are given regarding the use of data from tagged animals, which are directly applicable for researches using similar systems. However, the manuscript needs some major adjustments: reconsidering the methods used for temporal data and rewriting the discussion. Several statistical choices need more justification, references and explanation or it should be reconsidered if other analyses would be more appropriate. The discussion needs more focus, which should frame the focal study better in what it is: a case study (n=13) that looks at the habituation of loggers to common vampire bats. Habituation is only one of several possible tag effects, this should be made more clear in a focused discussion.

Please do not feel demotivated with the major and many comments I have. This is important data that should be published and I know too many stories about such data ending up on the shelf while it is necessary to publish these kind of case studies to push the field of animal-tagging forward (see Appendix A).

Decision letter (RSOS-211249.R0)

Dear Dr Carter

The Editors assigned to your paper RSOS-211249 "Habituation of common vampire bats to biologgers" have now received comments from reviewers and would like you to revise the paper in accordance with the reviewer comments and any comments from the Editors. Please note this decision does not guarantee eventual acceptance.

Please submit your revised manuscript and required files (see below) no later than 21 days from today's (ie 02-Sep-2021) date. Note: the ScholarOne system will 'lock' if submission of the revision is attempted 21 or more days after the deadline. If you do not think you will be able to meet this deadline please contact the editorial office immediately.

on behalf of Dr Joachim Mergeay (Associate Editor) and Kevin Padian (Subject Editor)
openscience@royalsociety.org

Associate Editor Comments to Author (Dr Joachim Mergeay):
Associate Editor: 1
Comments to the Author:
Dear authors,

We now have received two critical reviews of your manuscript. Both reviewers agree that this study is relevant in filling gaps in the effects of scientific tagging on animal behaviour, but also identify a number of weak points in the type of data presented, how the data are analysed and also highlight a lack of clarity in parts of the methods. I suggest you have thorough look at their comments and evaluate whether or not you can adapt your manuscript to meet their requirements.

Best regards,
Editor comments:
Thanks for your submission. As you will see, one reviewer recommended "reject" but their comments are not consistent so I think they may have meant "revise," and this is what we would like you to do. Please take all comments into account. Best wishes.

Reviewer comments to Author:

Reviewer: 1

Comments to the Author(s)

Comments on the manuscript entitled: Habituation of common vampire bats to biologgers

In the well-written manuscript, the authors present a test of habituation to tagging in common vampire bats. They assessed the tagging effect by video recording grooming behaviour shortly after the tag deployment. Overall, I like the idea of bridging the short- (rarely studied) and long-term (well studied) effects of biologging. Their results provide a valuable first insight into the process of habituation and might be helpful to researchers planning to start deploying tags to bats. I provide a couple of comments/suggestions that might improve the current version of the manuscript.

Main comments:

- 1) Authors used one type of tag on one species with very specific behaviour using rather low number of wild-caught individuals kept in cages. All of these aspects might have effects on the recorded behaviour and further generalization of the results in any research. However, any of these crucial aspects limiting generalization of manuscript results is neither discussed in the discussion nor mentioned anywhere in the manuscript. I would suggest to properly acknowledge these circumstances at least in the Discussion section.
- 2) Please specify conditions (L103-110) under which you kept bats after the bats were captured and before the bats were tested (approx. 1 month). Did you keep them in the small plastic boxes?
- 3) You state (L96) the bats were caged in small (28×28×40 cm) plastic boxes. I assume that bats could not fly in such small boxes and their behaviour was thus limited. This is another factor that could affect the tagging effects as bats could not behave naturally but is not discussed in the manuscript.

####

Minor comments:

L29: Could you specify (daytime/night) period of the day when individual behaviour was recorded? Was the behaviour recorded continuously for 12 hours, or over the 12-hour period in total?

L33: Please state sample sizes for these results if different from number of individuals used in the study (please see comment to L123-124 below).

L38: What does "solitary" means in this sentence? Is it related to species name, caging of individuals, something else?

L41: I would suggest to replace "remove" with "fall off".

L70-71: I would say that tags that are attached to animal's body with a harness can cause irritation too.

L79: You stated that 13 individuals were tested multiple times (L25, L103, Figure 3).

L122: What is a "flight cage"? This was not described yet.

L123-124: Therefore, short-term habituation could be assessed in nine individuals (assuming you had 13 individuals, not 14). This should be clearly stated throughout the manuscript.

L132: Please provide sample size for each result you provide - this will (i) allow easy interpretation of your results and (ii) allow incorporation of your results into future meta-analysis summarizing tagging effects across species.

L137: Do you really present relative measures of grooming? It seems to me you present only raw measures of tag- and self-grooming times.

L204: Did you mean "tracking" instead of "tacking"?

Figure 1: Please remove lines connecting measures within the 12-hour periods as no values have been measured during these periods.

L218-219: The effect of glue types was not tested in this study.

L230-240: I would suggest to remove the part of the text where authors elaborate on the anecdotal observations that are out of scope of their study.

Reviewer: 2

Comments to the Author(s)

The manuscript is an interesting contribution to the literature regarding the effects of animal-tagging. Some recommendations are given regarding the use of data from tagged animals, which are directly applicable for researches using similar systems. However, the manuscript needs some major adjustments: reconsidering the methods used for temporal data and rewriting the discussion. Several statistical choices need more justification, references and explanation or it should be reconsidered if other analyses would be more appropriate. The discussion needs more focus, which should frame the focal study better in what it is: a case study (n=13) that looks at the habituation of loggers to common vampire bats. Habituation is only one of several possible tag effects, this should be made more clear in a focused discussion.

Please do not feel demotivated with the major and many comments I have. This is important data that should be published and I know too many stories about such data ending up on the shelf while it is necessary to publish these kind of case studies to push the field of animal-tagging forward.

===PREPARING YOUR MANUSCRIPT===

===PREPARING YOUR REVISION IN SCHOLARONE===

Author's Response to Decision Letter for (RSOS-211249.R0)

See Appendix B.

Decision letter (RSOS-211249.R1)

Dear Dr Carter,

It is a pleasure to accept your manuscript entitled "Habituation of common vampire bats to biologgers" in its current form for publication in Royal Society Open Science. The comments of the reviewer(s) who reviewed your manuscript are included at the foot of this letter.

Kind regards,
Royal Society Open Science Editorial Office

on behalf of Dr Joachim Mergeay (Associate Editor) and Kevin Padian (Subject Editor)
openscience@royalsociety.org

Associate Editor Comments to Author (Dr Joachim Mergeay):

Comments to the Author:

Dear authors,

thank you for the revision, which seems satisfactory to me.
Well done.

Joachim Mergeay, associate editor

Appendix A

The manuscript is an interesting contribution to the literature regarding the effects of animal-tagging. Some recommendations are given regarding the use of data from tagged animals, which are directly applicable for researches using similar systems. However, the manuscript needs some major adjustments: reconsidering the methods used for temporal data and rewriting the discussion. Several statistical choices need more justification, references and explanation or it should be reconsidered if other analyses would be more appropriate. The discussion needs more focus, which should frame the focal study better in what it is: a case study (n=13) that looks at the habituation of loggers to common vampire bats. Habituation is only one of several possible tag effects, this should be made more clear in a focused discussion.

Please do not feel demotivated with the major and many comments I have. This is important data that should be published and I know too many stories about such data ending up on the shelf while it is necessary to publish these kind of case studies to push the field of animal-tagging forward.

Major comments:

I have some major concerns regarding the **methods** used. First, in my opinion, the study does not include an actual **control**. The researchers mainly compare tag-grooming with self-grooming, while the bats were still tagged. A more suitable control would be present in a before-after design, or even better, with a BACI (before after, control impact) design (Smith, 2002; Christie *et al.*, 2019) (a fully crossed design, with the term borrowed from environmental impact assessment studies). However, the latter would require a much larger sample size, which is probably not feasible with the focal study system. Behavioral assessments before the tagging of each individual would have been a much more suitable control. The lack of such a control should be mentioned. Second, **temporal data** are handled in quite a simplistic manner (rounding to nearest hour, default fitted lines with ggplot2). Although I am not familiar myself with how such data should properly be handled, there does not seem to be any robust statistical output present and the analyses are not supported by any literature. The **rounding to the hour** also needs some more explanation (what is zero? Is this a full hour of observations or only half an hour?) and sensitivity exploration (15min, 30min as bins?).

The **discussion** is currently chaotically written and several topics unnecessarily repeated. It could be writing much more to the point and straightforward if topics are handled separately in each paragraph. An essential topic missing in the paragraph is explanation regarding other types of tag effects and how the one studied here fits within these (habituation to tags is a very specific tag effect which is not directly comparable to for instance effect on flight performance or more long-term effects). The discussion should also be tailored more to what the study is about. For instance, the last paragraph is not per se relevant to the study here; certainly because in the focal study, effect sizes are not measured and are also not relevant for type of tag-effect measured; then claiming that effect sizes should be measured in future studies is off-topic. Another example, only in the discussion it is mentioned that there was apparently another objective of the study: look at the suitability of certain glues: this should be added to the objectives in the introduction (and methods & results) or minimized in the discussion (and not considered one of the major conclusions). Because the discussion is often off-topic, it gives the impression of overselling the study, while a more honest focus regarding the specific type of tag-effect used and framing it as a case-study (as still only 13 bats were assessed and often data was missing because bats were not in the frame of the camera or data was lost), would make it a much stronger manuscript.

More minor comments:

Title: to include the actual results of the study in the title, I would suggest to add 'after X hours' in the title (but this is of course also a matter of taste).

L56-67: What is explained in the examples is contradictory to the first sentence of the paragraph (that such studies tend to focus on long-term survival and reproductive success, rather than short-term effects). When a meta-analysis can already summarize information about foraging trip duration of 37 bird tagging studies, this seems already like a lot of studies to me. The other studies mentioned are also convincing examples of short-term tag effects. I also know a few more, including studies with insects (Boiteau & Colpitts, 2001; Wilson *et al.*, 2004; Boiteau *et al.*, 2010; Kim *et al.*, 2016). So the list of short-term impact assessments seems relatively long to me. The first sentence should be reformulated to better fit this notion and the examples given.

L63: it is bottlenose dolphin instead of 'bottle-nosed' dolphin. And please provide scientific names when mentioning species names (in this paragraph and further).

L62: omit 'effect size: fisher-transformed correlation coefficients =0.13'. These are details that can be looked-up in the reference given. This is also non-informative with no context of that study.

L67: dot missing.

L68-78: please provide references in this paragraph for the postulated assumptions. There are enough review-papers that have raised such concerns earlier (such as the references given on L54-55).

L79: here 14 individuals are mentioned, while later 13. Make sure this is consistent.

L89: "to test effects of proximity tags": be more specific. Throughout the manuscript you are very nuanced about what kind of effect you are exactly looking at (the behavioral habituation to tag attachment). Please also be more specific in this sentence.

L89-91: can you provide references if these tags were previously used in other studies (I think they are mentioned later on or earlier, such as Ripperger et al 2019, 2020)? It gives a good impression if you already show here that it is an established method or has been used in the past for behavioral studies.

L93: same as previous comment, because the sentence says 'typical way': this gives me an indication that this is an established method, so please give references for this postulated method/handling type.

L103: see previous comment, earlier 14 individuals were mentioned, here 13.

L104: here, Panamá is written with an accent, in the subsequent line without. Please be consistent throughout the manuscript.

L112-119: can you also provide references in this paragraph? Is this type of analysis an established method for this kind of data? If so, provide references.

L116: what is the influence of taking the mean for the nearest hour? Differently put, what is the influence on the results when looking at different timescales? Thus, what are the results when you would do this for 15min, 20min or half an hour? This is the type of questions that references to such established methods would avoid.

L119: please give references of the R packages whenever you use them. See <https://www.r-bloggers.com/2018/08/how-to-cite-packages/> do this throughout the manuscript. When the R package is loaded in R, you can do this by typing citation("boot"), this will give you the reference that should be cited.

L123: 4 out of how many? In total or of time block 2? Thank you for the transparent honesty about the missing data.

L125: replace 'her' by 'the'. The sex of the animal does not matter here. Try to avoid any gender-references in animal related research (unless it is of importance to the analysis/results), to make the writing as objective and descriptive as possible.

L133-L134: Because you work with the nearest hour, the formulation of 'after the first four hours' or 'during the first hour' is not really clear to me. After the first four hours, is this from hour 4 onwards? How do you define hour zero? Is 'during the first hour': is this hour 1 compared to hour 2? Solve this by a more consequent use of calibration zero-point and how hours are defined and used in the text. Make sure an equal amount of observations are assigned to the first bin (and not only half an hour for instance), and this should be clarified in the methods.

L135: it is not really clear to me how $n=54$ is established. If this is a mean of all hours after the first four, this does not seem like the proper statistical test to me to aggregate everything and just take the mean.

Figure 1: First, please remove the dot connections, these make the graph unnecessary busy and gives little, if any, extra information. Second, error bars are not visible although mentioned in the caption. I would rather see error bars (or shaded grey area, for example), than dot-connections. Third, there seems to be a lot of variation present in the self-grooming, this deserves a bit more explanation (in the discussion for instance). Probably, error bars would also clarify this?

Figure 2:

- please disconnect the letters and the numbers ('block1' should be 'block 1').
- I would omit the names given to the bats, this looks a bit unprofessional in a paper (I have also done this before, but that stayed in the raw dataset, and were replaced in the publication by numbering).
- I would recommend to number the individuals continually, not 'block 1' etc, because then someone needs to dive into the material and methods to understand this, while you want the figure to stand on its own as most people will look at the abstract and figures. I would number from 1 to 13 for individuals, but you could also number males and females, although nothing is done with the variable sex. So I would rather leave this variable for the raw data, but not include in the manuscript as nothing is done with it (and probably can't be done due to the low sample size).
- Please provide full axis numbering for x-axis, as only the first 5 hours are used, which won't make the axis too busy I think. Here as well (see previous comment), the calibration of the 0 is not clear to me (is this the first hour? Than the first 6 hours are shown instead of 5). Make sure that an equal number of observations is assigned to the first hour (I have the impression that it will only be the first half hour of data, but this should be clarified).
- 'Lines show expected rates': as this is a fitted line, formulate as 'fitted rates based on the data points' rather than 'expected'. I would use the term 'expected' if a predictive model (with variables explaining the data) was used to predict the grooming per hour, which is not the case here. Regarding this, I would combine the two first sentences, for instance: 'Curves are fitted rates of grooming directed at the proximity sensor tag (triangles and dashed line) versus rest of the body (circles and solid lines)'. I am also not sure if default fitted lines from the ggplot2 package are the preferable ones for temporal trends.

- Can you explain more 'that were observable within 30 min of tagging' within the caption? Why are not more individuals shown? If there are no data for some individuals during certain time frames, this could also be indicated in the graph (striped areas are grey areas for instance). I would prefer to see all individuals in this graph (as in figure 3). If only a few animals are given, it quickly feels like cherry-picking because these particular bats gave nice graphs.

- please also here give a reference if you mention the R package.

- please provide a legend within the figure itself for the symbols used.

Figure 3:

- See figure 2 comments regarding names.

- Same for 'expected'; reference to package.

- although 'triangles' and 'circles' are mentioned in the caption, these are not present on the graph and are quite important. Especially here, as no raw data is shown, I have my doubts about the suitability of default fitted lines of the ggplot2 package for temporal data. Can the usage of this be backed-up with literature and explained more (in material and methods for instance), or a more proper method used for temporal data? If these lines are only to 'guide the eye', this should be stated and explained.

- I would suggest to add a line in the two graphs with bats that removed their tag to indicate what timepoint exactly they removed their tag.

L171-173: in the figure, it is clear that a lot of variation is present, can the mean of self-grooming and CI be provided here? Otherwise not that quickly comparable to the examples given. And can you also find CI in the cited articles?

L172: "(e.g. 12 %; Stockmaier et al., 2018)" and "(e.g. 24 %; Wilkinson, 1986). I am also not sure why 'e.g.' is put? If you have more examples, please provide these so the comparison with the focal study can be made.

L172: be consistent if you put a space in between number and %-sign or not.

L175: increase (without s)

L178: were attached (instead of we attached).

L180: remove 'we' and make passive.

L183-185: there are also some studies looking at the use of glues in insects, which might be a good addition to the discussion, or just for your information: (Boiteau *et al.*, 2009; Lee *et al.*, 2013; Pope *et al.*, 2015)

L195: replace and by, for instance, 'however' in a new sentence (otherwise quite a long sentence, breaking it up and adding connecting words will make it easier to follow the reasoning).

L200-201: How did this exactly taught you to avoid a particular type of glue? Is this connected to the previous paragraph? This paragraph needs some rewriting to make the reasoning more clear.

L202: 'Another objective': this is the first time that another objective is mentioned. This is the actual main objective. I would say here 'the main objective', and rewrite the previous paragraph to indicate that the information there regarding glue is a side note (which does fit in the discussion, but should not be put forward as an objective).

L204: taking instead of tacking

Conclusions: this is as long as the discussion, and should actually be one small paragraph where the main findings are quickly rehearsed. No new discussion points or results should be given here.

L219: 'than' instead of at

L217-223: this paragraph needs more references regarding the adhesives.

L218-220: this is not a result from the material and methods or results sections. Please add there or omit this in the conclusions. It can be put somewhere in the discussion, but not as a main point if it is not considered in the body of the manuscript.

L241: the magnitude *of* tagging effects.

L231: remove ', and' after bats. Otherwise, the reader connect immediately distinguish between what you have anecdotally observed to the next sentence explaining this.

L224-242: these is a paragraph that should be moved to the discussion. Social effects are a new discussion topic and should be handled as such.

L235-242: finding no effect is indeed hard, that is also the way statistics just work. Although the reasoning here is a bit long and clumsy and could be formulated much sharper. For instance, the explanation regarding zebra finches looks a bit contradictory to what the actual point is of the paragraph. Also name-dropping publication bias, while effects of this on meta-analyses can be manifold, makes the paragraph confusing. In my opinion, the point made in this paragraph is also not relevant for the focal study, as looking at habituation over time is not looking at actual effect sizes. As such, your methods are not easily transferable to studies looking at for instance the effect on flight speed or flight performance. The point made here is in my opinion too general and not something you are able to recommend from the focal study results. I would actually omit this part and end with a summary conclusion regarding your results.

Boiteau, G. & Colpitts, B.G. (2001) Electronic tags for the tracking of insects in flight: Effect of weight on flight performance of adult Colorado potato beetles. *Entomologia Experimentalis et Applicata*, **100**, 187–193.

Boiteau, G., Meloche, F., Vincent, C. & Leskey, T.C. (2009) Effectiveness of Glues Used for Harmonic Radar Tag Attachment and Impact on Survival and Behavior of Three Insect Pests. *Environmental Entomology*, **38**, 168–175.

Boiteau, G., Vincent, C., Meloche, F. & Leskey, T.C. (2010) Harmonic radar: assessing the impact of tag weight on walking activity of Colorado Potato Beetle, Plum Curculio, and Western Corn Rootworm. *Journal of Economic Entomology*, **103**, 63–69.

Christie, A.P., Amano, T., Martin, P.A., Shackelford, G.E., Simmons, B.I. & Sutherland, W.J. (2019) Simple study designs in ecology produce inaccurate estimates of biodiversity responses. *Journal of Applied Ecology*, **56**, 2742–2754.

Kim, J., Jung, M., Kim, H.G. & Lee, D.-H. (2016) Potential of harmonic radar system for use on five economically important insects: Radar tag attachment on insects and its impact on flight capacity. *Journal of Asia-Pacific Entomology*, **19**, 371–375.

Lee, D.-H., Wright, S.E., Boiteau, G., Vincent, C. & Leskey, T.C. (2013) Effectiveness of Glues for Harmonic Radar Tag Attachment on Halyomorpha halys (Hemiptera: Pentatomidae) and Their

Impact on Adult Survivorship and Mobility. *Environmental Entomology*, **42**, 515–523.

Pope, T., Gundalai, E., Elliott, L., Blackshaw, R., Hough, G., Wood, A., Bennison, J., Prince, G. & Chandler, D. (2015) Recording the movement of adult vine weevil within strawberry crops using radio frequency identification tags. *Journal of Berry Research*, **5**, 197–206.

Smith, E.P. (2002) *BACI design*. *Encyclopedia of Environmetrics* (ed. by A.H. El-Shaarawi) and W.W. Piegorsch), pp. 141–148. John Wiley & Sons, Ltd, Chichester.

Wilson, R.P., Kreye, J.M., Lucke, K. & Urquhart, H. (2004) Antennae on transmitters on penguins: balancing energy budgets on the high wire. *The Journal of Experimental Biology*, **207**, 2649–2662.

Appendix B

Response to reviewer comments

Author responses are in bold text

Associate Editor Comments to Author (Dr Joachim Mergeay):

Associate Editor: 1

Comments to the Author:

Dear authors,

We now have received two critical reviews of your manuscript.

Both reviewers agree that this study is relevant in filling gaps in the effects of scientific tagging on animal behaviour, but also identify a number of weak points in the type of data presented, how the data are analysed and also highlight a lack of clarity in parts of the methods.

I suggest you have thorough look at their comments and evaluate whether or not you can adapt your manuscript to meet their requirements.

best regards,

We thank the reviewers for their helpful comments which we feel greatly improved the manuscript.
--Kline, Ripperger, and Carter

Editor comments:

Thanks for your submission. As you will see, one reviewer recommended "reject" but their comments are not consistent so I think they may have meant "revise," and this is what we would like you to do. Please take all comments into account. Best wishes.

Reviewer comments to Author:

Reviewer: 1

Comments to the Author(s)

Comments on the manuscript entitled: Habituation of common vampire bats to biologgers

In the well-written manuscript, the authors present a test of habituation to tagging in common vampire bats. They assessed the tagging effect by video recording grooming behaviour shortly after the tag deployment. Overall, I like the idea of bridging the short- (rarely studied) and long-term (well studied) effects of biologging. Their results provide a valuable first insight into the process of habituation and might be helpful to researchers planning to start deploying tags to bats. I provide a couple of comments/suggestions that might improve the current version of the manuscript.

We are glad that the reviewers found the manuscript well-written and the results valuable.

Main comments:

1) Authors used one type of tag on one species with very specific behaviour using rather low number of wild-caught individuals kept in cages. All of these aspects might have effects on the recorded behaviour and further generalization of the results in any research. However, any of these crucial aspects limiting generalization of manuscript results is neither discussed in the discussion nor mentioned anywhere in the manuscript. I would suggest to properly acknowledge these circumstances at least in the Discussion section.

Thanks for this feedback. We added the following to the second paragraph of the discussion:

"It is important to note the limitations of this study. We measured a single behavior in a few isolated individuals. Although we compare tag-grooming with self-grooming as control measure, we do not have a control period prior to the application of the tag. We also cannot know how

these tags affect behaviors such as flight performance or social behavior. One should therefore be cautious overgeneralizing from these findings. However, some observations are noteworthy.”

2) Please specify conditions (L103-110) under which you kept bats after the bats were captured and before the bats were tested (approx. 1 month). Did you keep them in the small plastic boxes?

Fixed. We added here:

“Before the bats were tagged and housed in clear plastic cages, they were housed communally in a flight cage (either 2.1 x 1.7 x 2.3 m or ~ 1.7 x 1 x 2.3 m).”

3) You state (L96) the bats were caged in small (28x28x40 cm) plastic boxes. I assume that bats could not fly in such small boxes and their behaviour was thus limited. This is another factor that could affect the tagging effects as bats could not behave naturally but is not discussed in the manuscript.

Yes, but the vast majority of social interactions among vampire bats occur within a roost which can be a tight cavity. We added:

“Although the bats could not fly, our goal here was to measure how the tags would affect the behavior of roosting bats.”

####

Minor comments:

L29: Could you specify (daytime/night) period of the day when individual behaviour was recorded? Was the behaviour recorded continuously for 12 hours, or over the 12-hour period in total?

Fixed. “Changed to 1800 to 0600 h”

L33: Please state sample sizes for these results if different from number of individuals used in the study (please see comment to L123-124 below).

Fixed.

L38: What does “solitary” means in this sentence? Is it related to species name, caging of individuals, something else?

Changed to “isolated individual vampire bats...”

L41: I would suggest to replace “remove” with “fall off”.

Fixed.

L70-71: I would say that tags that are attached to animal's body with a harness can cause irritation too.

We changed “glued” to “attached”.

L79: You stated that 13 individuals were tested multiple times (L25, L103, Figure 3).

To reduce redundancy, we deleted this number in the introduction.

L122: What is a “flight cage”? This was not described yet.

For clarity, changed to “room”

L123-124: Therefore, short-term habituation could be assessed in nine individuals (assuming you had 13 individuals, not 14). This should be clearly stated throughout the manuscript.

Fixed. We now state the number of bats used for each estimate:

“During the first hour, the mean tag-grooming rate declined dramatically from 53% (95% CI = 36--65 %, n=6 bats) to 16% (8--24%, n=9 bats; Figure 1).”

L132: Please provide sample size for each result you provide – this will (i) allow easy interpretation of your results and (ii) allow incorporation of your results into future meta-analysis summarizing tagging effects across species.

Fixed. See above.

L137: Do you really present relative measures of grooming? It seems to me you present only raw measures of tag- and self-grooming times.

For clarity, we revised this sentence to read:

“The habituation was evident in tag-grooming rates in comparison to self-grooming rates within the first hour across all six bats that were observable during that time (Figure 2).”

L204: Did you mean “tracking” instead of “tacking”?

Yes, fixed.

Figure 1: Please remove lines connecting measures within the 12-hour periods as no values have been measured during these periods.

Fixed.

L218-219: The effect of glue types was not tested in this study.

We agree. We cut this text.

L230-240: I would suggest to remove the part of the text where authors elaborate on the anecdotal observations that are out of scope of their study.

As suggested, we cut this text considerably:

Original:

“Social effects are likely to vary greatly by species and behavior; captive African elephants showed no changes in social behavior when wearing tracking collars (Horback et al., 2012), but free-ranging greater spear-nosed bats showed increased aggression towards individuals with glowing tags compared to untagged control bats, possibly due to neophobia or light aversion (Hoxeng et al., 2007). Anecdotally, we have often observed vampire bats grooming the tags or metal bat bands on other bats, and social grooming increases immediately after (1) the actor was self-grooming, (2) the recipient was self-grooming, or (3) the recipient has wetted or disturbed fur (Narizano & Carter, 2020). If proximity sensors alter association rates among pairs, then the resulting social networks might vary from the typical networks experienced by untagged bats. However, significant effects of tags are not guaranteed. To take one example: despite widespread belief and several studies reporting that colored bands affect the mating behavior and fitness of zebra finches, these conclusions appear to be due largely or completely to publication bias against null findings, and the real effect size of color bands on zebra finch fitness is essentially zero (Wang et al., 2018). It is always most useful to *measure* the magnitude tagging effects on social behavior, rather than merely testing the null hypothesis of no effect.”

Revised:

“Social effects are likely to vary greatly by species and behavior; captive African elephants showed no changes in social behavior when wearing tracking collars (Horback et al., 2012), but free-ranging greater spear-nosed bats showed increased aggression towards individuals with glowing tags compared to untagged control bats, possibly due to neophobia or light aversion (Hoxeng et al., 2007). If proximity sensors alter substantially association rates among tagged pairs for a period after attachment, then this should be taken into account when constructing social networks.”

Reviewer: 2

Comments to the Author(s)

The manuscript is an interesting contribution to the literature regarding the effects of animal-tagging. Some recommendations are given regarding the use of data from tagged animals, which are directly applicable for researches using similar systems. However, the manuscript needs some major adjustments: reconsidering the methods used for temporal data and rewriting the discussion. Several statistical choices need more justification, references and explanation or it should be reconsidered if other analyses would be more appropriate. The discussion needs more focus, which should frame the focal study better in what it is: a case study (n=13) that looks at the habituation of loggers to common vampire bats. Habituation is only one of several possible tag effects, this should be made more clear in a focused discussion.

Please do not feel demotivated with the major and many comments I have. This is important data that should be published and I know too many stories about such data ending up on the shelf while it is necessary to publish these kind of case studies to push the field of animal-tagging forward.

Major comments:

I have some major concerns regarding the **methods** used. First, in my opinion, the study does not include an actual **control**. The researchers mainly compare tag-grooming with self-grooming, while the bats were still tagged. A more suitable control would be present in an before-after design, or even better, with a BACI (before after, control impact) design (Smith, 2002; Christie *et al.*, 2019) (a fully crossed design, with the term borrowed from environmental impact assessment studies). However, the latter would require a much larger sample size, which is probably not feasible with the focal study system. Behavioral assessments before the tagging of each individual would have been a much more suitable control. The lack of such a control should be mentioned.

The discussion now has the following text:

“It is important to note the limitations of this study. We measured a single behavior in a few isolated individuals. Although we compare tag-grooming with self-grooming as control measure, we do not have a control period prior to the application of the tag. We also cannot know how these tags affect behaviors such as flight performance or social behavior. One should therefore be cautious overgeneralizing from these findings.”

Second, **temporal data** are handled in quite a simplistic manner (rounding to nearest hour, default fitted lines with ggplot2). Although I am not familiar myself with how such data should properly be handled, there does not seem to be any robust statistical output present and the analyses are not supported by any literature. The **rounding to the hour** also needs some more explanation (what is zero? Is this a full hour of observations or only half an hour?) and sensitivity exploration (15min, 30min as bins?).

Our question is simple (how do the grooming rates towards the tag and body change over time?), and we believe our plots and analyses are correspondingly simple, although not simplistic, because they are not misleadingly simple. We think that the best way to capture the habituation curve is to show it visually, and we presented three different visualizations. Figure 1 shows the mean rates for each hour with 95% bootstrapped CIs for the three days to show the uncertainty around the estimates. Figures 2 and 3 show the first five hours with local polynomial regression curves to describe the overall change over time during the period where the bats habituated. We also publish the raw data so readers can plot the data differently. The comparison of mean values and their bootstrapped 95% confidence intervals is standard practice and doesn't require justification in the literature.

We did not actually use the default fitted lines (as believed by the reviewer); we considered these to be the most appropriate lines for describing a non-linear curve. Calculating a rate requires binning the data in time, so we rounded time to the nearest hour which makes visualization of the raw data easier than when all data points are 0 or 1. For clarity, we added the following underlined text in the methods:

“We rounded the time of observations to the nearest hour (e.g. 0 hours is time 0 to 00:29:59, hour 1 is 00:30:00 to 1:29:59), then used nonparametric bootstrapping to generate a 95% confidence interval around the mean rate of self-grooming and tag-grooming for every hour after attachment.”

The **discussion** is currently chaotically written and several topics unnecessarily repeated. It could be writing much more to the point and straightforward if topics are handled separately in each paragraph. An essential topic missing in the paragraph is explanation regarding other types of tag effects and how the one studied here fits within these (habituation to tags is a very specific tag effect which is not directly comparable to for instance effect on flight performance or more long-term effects).

We thank the reviewer for pointing this out. We have greatly reduced and revised the entire discussion to remove redundancy, and we now discuss habituation as only one possible behavioral response.

The much shorter discussion now reads:

“Solitary tagged vampire bats habituated to glued-on tracking devices within a few hours as long as the device remained securely attached. The bats spent more than half their time manipulating the tag during the first half hour, but by the fifth hour, tag grooming rates dropped to an average of 4% and were lower than self-grooming rates (directed to the bat's body). Self-grooming rates remained stable over time (Figure 1; mean = 16%; 95% CI = 14 to 18%, range = 0 to 53%; n = 58 hours), and were comparable with estimates from other captive studies (12%, Stockmaier et al., 2018) and field observations (24%, Wilkinson, 1986).

It is important to note the limitations of this study. We measured a single behavior in a few isolated individuals. Although we compare tag-grooming with self-grooming as control measure, we do not have a control period prior to the application of the tag. We also cannot know how these tags affect behaviors such as flight performance or social behavior. One should therefore be cautious overgeneralizing from these findings. However, some observations are noteworthy. First, two bats were able to loosen the attachment of the tag which caused them to dishabituate and increase tag-grooming rates. The behavior of tagged animals might therefore be impacted both immediately after tag attachment and immediately before tags fall off. Second, these tags fell off much faster than in a previous study where all 50 proximity sensors that were attached to free-ranging vampire bats stayed attached for 9 days or longer (Ripperger et al., 2019). In this earlier study, we used a different surgical glue (the rubber-based Perma-Type Surgical Cement, Perma-

Type Company, Inc., Plainville, Connecticut) which is almost twice as strong as the latex-based surgical glue used here, Osto-Bond (Carter, Sichmeller & Hohmann, 2009). When using glue for tag attachment, we therefore recommend species-specific tests with different glues (Carter, Sichmeller & Hohmann, 2009) and reporting what glue is used.

An important next step is to evaluate to what extent animal-borne tags alter social behaviors. Social effects are likely to vary greatly by species and behavior; captive African elephants (*Loxodonta africana*) showed no changes in social behavior when wearing tracking collars (Horback et al., 2012), but free-ranging greater spear-nosed bats (*Phyllostomus hastatus*) showed increased aggression towards individuals with glowing tags compared to untagged control bats, possibly due to neophobia or light aversion (Hoxeng et al., 2007). If proximity sensors alter association rates among tagged pairs for a period after attachment, then this should be taken into account when constructing social networks.

There is increasing evidence across animal taxa that tagging can have considerable effects on survival, reproduction, parental care, or behavior (Bodey et al., 2018). In insectivorous bats that forage in flight, there is experimental evidence that tagged individuals experience a decline in maneuverability (Aldridge & Brigham, 1988), but no effects on reproduction or body mass from radio-tagging were detected in a long-term field study (Neubaum et al., 2005). Changes in body mass before and after tagging are common ways of evaluating possible tagging effects, but sample sizes are often small and hence incapable of detecting subtle effects (Roeleke et al., 2016; Voigt et al., 2020). Also, a sampling bias occurs if not all tagged animals are recaptured for measurement, because the animals that are most impacted might not be measured. For these reasons, captive measurements of tagging effects provide an important method for estimating behavioral responses, and are useful for biologging studies.

Conclusions

In solitary vampire bats, habituation to proximity sensors occurs within the first 1-3 hours of tag attachment—but only if the tag is securely attached. Attachment methods should therefore be carefully considered and tested. Our findings highlight the need for preliminary testing of biologgers and other tags with captive animals whenever feasible.

The discussion should also be tailored more to what the study is about. For instance, the last paragraph is not per se relevant to the study here; certainly because in the focal study, effect sizes are not measured and are also not relevant for type of tag-effect measured; then claiming that effect sizes should be measured in future studies is off-topic.

We have removed this section.

Another example, only in the discussion it is mentioned that there was apparently another objective of the study: look at the suitability of certain glues: this should be added to the objectives in the introduction (and methods & results) or minimized in the discussion (and not considered one of the major conclusions).

We removed this section, as it was not something we intended to study, but merely something we realized after the fact.

Because the discussion is often off-topic, it gives the impression of overselling the study, while a more honest focus regarding the specific type of tag- effect used and framing it as a case-study (as still only 13 bats were assessed and often data was missing because bats were not in the frame of the camera or data was lost), would make it a much stronger manuscript.

See revised discussion above.

More minor comments:

Title: to include the actual results of the study in the title, I would suggest to add 'after X hours' in the title (but this is of course also a matter of taste).

We decided to use the shorter title without the number of hours

L56-67: What is explained in the examples is contradictory to the first sentence of the paragraph (that such studies tend to focus on long-term survival and reproductive success, rather than short-term effects). When a meta-analysis can already summarize information about foraging trip duration of 37 bird tagging studies, this seems already like a lot of studies to me. The other studies mentioned are also convincing examples of short-term tag effects. I also know a few more, including studies with insects (Boiteau & Colpitts, 2001; Wilson *et al.*, 2004; Boiteau *et al.*, 2010; Kim *et al.*, 2016). So the list of short-term impact assessments seems relatively long to me. The first sentence should be reformulated to better fit this notion and the examples given.

Thanks for pointing this out. We revised this sentence and changed “only a few studies” to “several studies”.

L63: it is bottlenose dolphin instead of 'bottle-nosed' dolphin. And please provide scientific names when mentioning species names (in this paragraph and further).

Fixed

L62: omit 'effect size: fisher-transformed correlation coefficients =0.13'. These are details that can be looked-up in the reference given. This is also non-informative with no context of that study.

Fixed

L67: dot missing.

Fixed

L68-78: please provide references in this paragraph for the postulated assumptions. There are enough review-papers that have raised such concerns earlier (such as the references given on L54-55).

We removed this sentence, because it is unnecessary.

L79: here 14 individuals are mentioned, while later 13. Make sure this is consistent.

Fixed.

L89: “to test effects of proximity tags”: be more specific. Throughout the manuscript you are very nuanced about what kind of effect you are exactly looking at (the behavioral habituation to tag attachment). Please also be more specific in this sentence.

Changed to: “To test if and how bats habituate to attached proximity tags,”

L89-91: can you provide references if these tags were previously used in other studies (I think they are mentioned later on or earlier, such as Ripperger et al 2019, 2020)? It gives a good impression if you already show here that it is an established method or has been used in the past for behavioral studies.

Added

L93: same as previous comment, because the sentence says 'typical way': this gives me an indication that this is an established method, so please give references for this postulated method/handling type.

Added

L103: see previous comment, earlier 14 individuals were mentioned, here 13.

Fixed

L104: here, Panamá is written with an accent, in the subsequent line without. Please be consistent throughout the manuscript.

Fixed

L112-119: can you also provide references in this paragraph? Is this type of analysis an established method for this kind of data? If so, provide references.

We now provide three references including citing the R package.

L116: what is the influence of taking the mean for the nearest hour? Differently put, what is the influence on the results when looking at different timescales? Thus, what are the results when you would do this for 15min, 20min or half an hour? This is the type of questions that references to such established methods would avoid.

Here is no objective right answer for how to bin estimate of time to show in a plot. If you calculate rates for smaller time bins, the confidence intervals on the mean will get larger, but you will have more means side-by-side over time. If you use larger time bins, the CIs get smaller but you have fewer points over time. We provide the raw data and R code to plot the data differently.

L119: please give references of the R packages whenever you use them. See <https://www.r-bloggers.com/2018/08/how-to-cite-packages/> do this throughout the manuscript. When the R package is loaded in R, you can do this by typing citation("boot"), this will give you the reference that should be cited.

Fixed. See above.

L123: 4 out of how many? In total or of time block 2? Thank you for the transparent honesty about the missing data.

Fixed.

L125: replace 'her' by 'the'. The sex of the animal does not matter here. Try to avoid any gender-references in animal related research (unless it is of importance to the analysis/results), to make the writing as objective and descriptive as possible.

Fixed

L133-L134: Because you work with the nearest hour, the formulation of 'after the first four hours' or 'during the first hour' is not really clear to me. After the first four hours, is this from hour 4 onwards? How

do you define hour zero? Is 'during the first hour': is this hour 1 compared to hour 2? Solve this by a more consequent use of calibration zero-point and how hours are defined and used in the text. Make sure an equal amount of observations are assigned to the first bin (and not only half an hour for instance), and this should be clarified in the methods.

We clarified this in the revision above. The rate for hour 0 is minutes 0 to 29, the rate for hour 1 is minutes 30 to 59 and so on. This means that the first rate of grooming for hour zero is based on 30 samples rather than 60, but all rates of grooming are percentages from dividing by the possible observations of grooming, so they are not biased by the number of observations per time bin. For example, some samples have fewer possible observations because the bat might have been off camera. The binning of the rate estimates cannot generate the effect we observed of the rates declining over time.

L135: it is not really clear to me how n=54 is established. If this is a mean of all hours after the first four, this does not seem like the proper statistical test to me to aggregate everything and just take the mean.

We revised this for clarity. To measure consistency over time, we report the mean, CI, and range for each hour for the rest of the study hours, not the mean across bats. We revised this text as follows:

“During the first hour, the mean tag-grooming rate declined dramatically from 53% (95% CI = 36--65 %, n=6 bats) to 16% (8--24%, n=9 bats; Figure 1). From hours 4 to 79, the mean tag-grooming rate ranged from 0 to 9.3% (mean = 3.6%, 95% CI = 3.0--4.2%, n= 54 hours, Figure 1). During this same period of time, mean grooming rates directed to other places on the bat’s body did not decline (Figure 1).”

Figure 1: First, please remove the dot connections, these make the graph unnecessary busy and gives little, if any, extra information. Second, error bars are not visible although mentioned in the caption. I would rather see error bars (or shaded grey area, for example), than dot-connections. Third, there seems to be a lot of variation present in the self-grooming, this deserves a bit more explanation (in the discussion for instance). Probably, error bars would also clarify this?

The plot may not have rendered correctly. We will check the PDF in the next version. In our submitted version, the Figure 1 does have errors bars (95% CIs, see below). We believe the lines between the points are necessary to clearly see the variation and trend over time.

Figure 2:
- please disconnect the letters and the numbers ('block1' should be 'block 1').

Fixed

- I would omit the names given to the bats, this looks a bit unprofessional in a paper (I have also done this before, but that stayed in the raw dataset, and were replaced in the publication by numbering).

Fixed. We now label the bats as Male 1, Male 2 etc

- I would recommend to number the individuals continually, not 'block 1' etc, because then someone needs to dive into the material and methods to understand this, while you want the figure to stand on its own as most people will look at the abstract and figures. I would number from 1 to 13 for individuals, but you could also number males and females, although nothing is done with the variable sex. So I would rather leave this variable for the raw data, but not include in the manuscript as nothing is done with it (and probably can't be done due to the low sample size).

We left the sex labels so that readers can see the lack of any difference by sex, so the bats are labeled Male 1-5 and Female 1-8. We don't believe this will confuse any readers.

- Please provide full axis numbering for x-axis, as only the first 5 hours are used, which won't make the axis too busy I think. Here as well (see previous comment), the calibration of the 0 is not clear to me (is this the first hour? Than the first 6 hours are shown instead of 5). Make sure that an equal number of observations is assigned to the first hour (I have the impression that it will only be the first half hour of data, but this should be clarified).

Fixed We now label every hour mark. We already explained the meaning of zero hour bin above.

- 'Lines show expected rates': as this is a fitted line, formulate as 'fitted rates based on the data points' rather than 'expected'. I would use the term 'expected' if a predictive model (with variables explaining the data) was used to predict the grooming per hour, which is not the case here. Regarding this, I would combine the two first sentences, for instance: 'Curves are fitted rates of grooming directed at the proximity sensor tag (triangles and dashed line) versus rest of the body (circles and solid lines)'. I am also not sure if default fitted lines from the ggplot2 package are the preferable ones for temporal trends.

Fixed. We did not use the default for the geom_smooth() function in ggplot; we set the span parameter to 2. LOESS (locally weighted smoothing) is a commonly preferred tool for temporal trends.

- Can you explain more 'that were observable within 30 min of tagging' within the caption? Why are not more individuals shown? If there are no data for some individuals during certain time frames, this could also be indicated in the graph (striped areas are grey areas for instance). I would prefer to see all individuals in this graph (as in figure 3). If only a few animals are given, it quickly feels like cherry-picking because these particular bats gave nice graphs.

We changed the figure caption title to "Habituation during first five hours for the six bats that were observable on camera within 30 min of tagging". The reader can see the curves for all 13 bats in Figure 3. We selected these individuals because they were visible on camera during the 30-60 min where most habituation takes place . The other bats cannot produce habituation curves for that period of time because they were not visible. Again, all the data are attached as a supplement for further analysis.

- please also here give a reference if you mention the R package

Fixed

- please provide a legend within the figure itself for the symbols used.

To avoid redundancy and save space, we chose to keep the legend in the caption.

Figure 3:

- See figure 2 comments regarding names.
- Same for 'expected'; reference to package.

Fixed

- although 'triangles' and 'circles' are mentioned in the caption, these are not present on the graph and are quite important. Especially here, as no raw data is shown, I have my doubts about the suitability of default fitted lines of the ggplot2 package for temporal data. Can the usage of this be backed-up with literature and explained more (in material and methods for instance), or a more proper method used for temporal data? If these lines are only to 'guide the eye', this should be stated and explained.

We have now added the raw data to the plot.

We use loess smoothing which is probably the most common way to show trends for noisy temporal data. We now cite: Cleveland WS, Devlin SJ. 1988. Locally Weighted Regression: An Approach to Regression Analysis by Local Fitting. *Journal of the American Statistical Association* 83:596–610. DOI: 10.2307/2289282.

- I would suggest to add a line in the two graphs with bats that removed their tag to indicate what timepoint exactly they removed their tag.

We now clarify that they removed their tags “where the observations end.”

L171-173: in the figure, it is clear that a lot of variation is present, can the mean of self-grooming and CI be provided here? Otherwise not that quickly comparable to the examples given. And can you also find CI in the cited articles?

“Self-grooming rates remained stable over time (Figure 1; mean = 16 %; 95% CI = 14—18 %, range = 0 to 53%; n= 58 hours), and were comparable with estimates from other captive studies (12 % (Stockmaier et al., 2018)) and field observations (24%, (Wilkinson, 1986)).”

L172: “(e.g. 12 %; Stockmaier et al., 2018)” and “(e.g. 24 %; Wilkinson, 1986). I am also not sure why 'e.g.' is put? If you have more examples, please provide these so the comparison with the focal study can be made.

We removed “e.g.”

L172: be consistent if you put a space in between number and %-sign or not.

Fixed throughout

L175: increase (without s)

Fixed

L178: were attached (instead of we attached).

Fixed

L180: remove 'we' and make passive.

Changed

L183-185: there are also some studies looking at the use of glues in insects, which might be a good addition to the discussion, or just for your information: (Boiteau *et al.*, 2009; Lee *et al.*, 2013; Pope *et al.*, 2015)

Thanks for this information.

L195: replace and by, for instance, 'however' in a new sentence (otherwise quite a long sentence, breaking it up and adding connecting words will make it easier to follow the reasoning).

Corrected. We rewrote this section. See above

L200-201: How did this exactly taught you to avoid a particular type of glue? Is this connected to the previous paragraph? This paragraph needs some rewriting to make the reasoning more clear.

We cut this text.

L202: 'Another objective': this is the first time that another objective is mentioned. This is the actual main objective. I would say here 'the main objective', and rewrite the previous paragraph to indicate that the information there regarding glue is a side note (which does fit in the discussion, but should not be put forward as an objective).

We rewrote this section. See above.

L204: taking instead of tacking

We cut this text.

Conclusions: this is as long as the discussion, and should actually be one small paragraph where the main findings are quickly rehearsed. No new discussion points or results should be given here.

We rewrote this section. See above.

L219: 'than' instead of at

We cut this text.

L217-223: this paragraph needs more references regarding the adhesives.

We cut this text.

L218-220: this is not a result from the material and methods or results sections. Please add there or omit this in the conclusions. It can be put somewhere in the discussion, but not as a main point if it is not considered in the body of the manuscript.

We cut this text.

L241: the magnitude *of* tagging effects.

We cut this text.

L231: remove ', and' after bats. Otherwise, the reader connect immediately distinguish between what you have anecdotally observed to the next sentence explaining this.

We cut this text.

L224-242: these is a paragraph that should be moved to the discussion. Social effects are a new discussion topic and should be handled as such.

Fixed

L235-242: finding no effect is indeed hard, that is also the way statistics just work. Although the reasoning here is a bit long and clumsy and could be formulated much sharper. For instance, the explanation regarding zebra finches looks a bit contradictory to what the actual point is of the paragraph. Also name-dropping publication bias, while effects of this on meta-analyses can be manifold, makes the paragraph confusing. In my opinion, the point made in this paragraph is also not relevant for the focal study, as looking at habituation over time is not looking at actual effect sizes. As such, your methods are not easily transferable to studies looking at for instance the effect on flight speed or flight performance. The point made here is in my opinion too general and not something you are able to recommend from the focal study results. I would actually omit this part and end with a summary conclusion regarding your results.

Agreed, we followed these recommendations and cut this text down. See new discussion above.

We thank both reviewers for their extensive and constructive comments, which have improved the clarity, brevity, and presentation of this manuscript.